# Real-World Effectiveness and Safety of Direct-Acting Antivirals in Patients with Chronic Hepatitis C and Epilepsy: An Epi-Ter-2 Study in Poland

**DOI:** 10.3390/jpm13071111

**Published:** 2023-07-09

**Authors:** Monika Pazgan-Simon, Jerzy Jaroszewicz, Krzysztof Simon, Beata Lorenc, Marek Sitko, Dorota Zarębska-Michaluk, Dorota Dybowska, Magdalena Tudrujek-Zdunek, Hanna Berak, Włodzimierz Mazur, Jakub Klapaczyński, Ewa Janczewska, Anna Parfieniuk-Kowerda, Robert Flisiak

**Affiliations:** 1Department of Infectious Diseases, Regional Specialistic Hospital, 50-149 Wrocław, Poland; krzysimon@gmail.com; 2Department of Infectious Diseases and Hepatology, Medical University of Silesia in Katowice, 41-902 Bytom, Poland; jerzy.jr@gmail.com; 3Department of Infectious Diseases and Hepatology, Wroclaw Medical University, 51-149 Wrocław, Poland; 4Department of Infectious Diseases, Pomeranian Center of Infectious Diseases, University of Gdansk, 80-210 Gdańsk, Poland; lormar@gumed.edu.pl; 5Department of Infectious and Tropical Diseases, Collegium Medicum, Jagiellonian University, 30-252 Kraków, Poland; sitkomar@o2.pl; 6Department of Infectious Disease, Voivodeship Hospital, Jan Kochanowski University, 25-369 Kielce, Poland; dorota1010@tlen.pl; 7Department of Infectious Diseases and Hepatology, Faculty of Medicine, Nicolaus Copernicus University, 87-100 Bydgoszcz, Poland; d.dybowska@wsoz.pl; 8Department of Infectious Diseases, Medical University of Lublin, 20-059 Lublin, Poland; magdalena.tudrujek@gmail.com; 9Hospital for Infectious Diseases, Warsaw Medical University, 02-091 Warszawa, Poland; hberak@wp.pl; 10Clinical Department of Infectious Diseases, Medical University of Silesia, 40-055 Chorzów, Poland; wlodek.maz@gmail.com; 11Department of Internal Medicine and Hepatology, Central Clinical Hospital of Internal Affairs and Administration, 02-241 Warszawa, Poland; klapaj@gmail.com; 12Department of Basic Medical Sciences, School of Public Health in Bytom, Medical University of Silesia, 40-055 Bytom, Poland; e.janczewska@poczta.fm; 13ID Clinic, Hepatology Outpatient Department, 41-400 Mysłowice, Poland; 14Department of Infectious Diseases and Hepatology, Medical University of Białystok, 15-540 Białystok, Poland; anna.parfieniuk@gmail.com (A.P.-K.); robert.flisiak@umb.edu.pl (R.F.)

**Keywords:** HCV infection, epilepsy, DAA treatment

## Abstract

Introduction: In Poland, active HCV infection affects between 0.4 and 0.5% of the population, i.e., about 150,000 people, while the number of patients with epilepsy is estimated to be 350,000–400,000. Currently available antiviral therapies show little interaction with neurological drugs. The aim of our study was to evaluate the effectiveness and safety of the treatment of chronic HCV infection in patients with coexisting epilepsy. Methods: A total of 184 epilepsy patients were selected from the group of 10,152 HCV-infected patients treated for HCV infection within the Epiter-2 database from 2015 to 2018. Comparing the effectiveness and safety of anti-HCV regimens between the patients with comorbid epilepsy and 3573 patients without comorbidities was our study’s objective. Results: The effectiveness of anti-HCV treatment was high in both the sample and the control group. No statistically significant SVR difference was observed between the sample group, with ITT = 93.5% and mITT = 95.5%, and the control group, with ITT = 95.2% and mITT = 97.5%, regardless of the genotype and the stage of liver disease at the start of therapy. The treatment was safe in patients with epilepsy. Conclusions: The effectiveness and safety of HCV treatment in patients with epilepsy are comparable to those of patients with no significant comorbidities.

## 1. Introduction

The World Health Organization (WHO) estimates the global number of patients actively infected with HCV to be 71 million and the global number of patients with epilepsy to be 60–70 million [1,2]. In Poland, active HCV infection affects between 0.4 and 0.5% of the population, i.e., about 150,000 people [3], while the number of epileptic patients is estimated to be 350,000–400,000 people [4]. However, it is not known how many epilepsy patients—regardless of the type and form of treatment of the condition—are infected with HCV and require treatment only for that reason. Despite the wide availability of diagnostic tools and access to highly effective antiviral therapy (since 2016), currently based only on direct antiviral agents (DAAs), HCV remains one of the main factors responsible for chronic liver disease in many countries. It is estimated that up to 40% of acute HCV infections resolve spontaneously, while 20–25% of chronically infected patients develop liver cirrhosis within 20 years of infection [5]. The numerous factors accelerating the progression to cirrhosis include both those that depend on the virus (the viral load and specific genotype) and those that depend on the host itself. These risk factors are male sex, prolonged infection, and coexisting liver diseases (fatty liver disease, toxic liver damage due to alcohol consumption, numerous drugs, coinfection with HBV or HIV, high BMI, and immunosuppression) [6]. With active HCV replication, approximately 5% of patients with cirrhosis developing within one year and 30% of patients with liver cirrhosis developing within ten years experience the decompensation of liver function. These consequences are portal hypertension and its complications, hepatic encephalopathy, cholestasis, hepatorenal syndrome, hepatopulmonary syndrome, hepatocardiac syndrome, cachexia, increased susceptibility to infections, and primary hepatocellular carcinoma—HCC (also mixed HCC/cHCC). On a global scale, liver failure results in 400,000 deaths annually. Moreover, numerous extrahepatic manifestations (EHM) of HCV infections, usually coexisting with liver disease, are observed in 30–40% of patients [7]. Virtually all organs and systems of the body are affected by this infection; the most common EHMs are mixed cryoglobulinemia (often asymptomatic) and non-Hodgkin’s B-cell lymphoma (B-NHL), but neurological symptoms also occur [8].

The treatment is aimed at eliminating the HCV infection, which allows it to inhibit or reverse the identified histological changes (especially less advanced changes); reduce the risk of developing liver cirrhosis, hepatocellular carcinoma, or EHMs; and eliminate the source of new infections [9,10]. However, unfortunately, even the permanent elimination of HCV does not lead to full recovery from cirrhosis, hepatocellular carcinoma, and many extrahepatic manifestations that may have already developed [11]. Hence, patients with HCV should be qualified for treatment as soon as possible. The decision to undertake therapy must take into account the perspective of patient cooperation and, when choosing a treatment regimen, the safety profile of DAAs in relation to the stage of liver disease and the presence of comorbidities and fitting pharmacotherapy methods. The patient should be informed about the necessity to follow the recommended treatment regimen and to monitor the course of therapy systematically, especially if other diseases are present. It is especially significant in patients with epilepsy, as the disease has various etiologies and is hence treated with different medications, which means interactions with anti-epileptic drugs are an actual possibility (Table 1). This is especially true in patients treated with carbamazepine, phenytoin, and phenobarbital due to the proven effects of these drugs on the metabolism of numerous DAAs [12,13]. This may result in a reduction in the concentration of DAA in the plasma and, consequently, lead to an ineffective antiviral treatment and inversely affect the epilepsy treatment [14].

The first non-interferon drug treating HCV infections was sofosbuvir, which was registered in 2014. Earlier therapies used interferon with the addition of DAAs or drugs with weaker virucidal potential, such as asunaprevir/daclatasvir. The currently available direct-acting antivirals (DAAs) act on known HCV replication targets, effectively leading to the elimination of the virus. The combinations currently used in therapy mainly include pangenotypic DAAs: sofosbuvir (NS5B polymerase inhibitor)/velpatasvir (NS5A inhibitor) and a variant with a third drug, voxilapreviu (NS3/4A inhibitor); ombitasvir (NS5A inhibitor)/paritaprevir (NS3/4A inhibitor); and ritonavir/dasabuvir (non-nucleotide NS5A inhibitor). Another current combination that is effective in treating infections with genotypes 1 and 4 is grazoprevir (NS3/4A protease inhibitor)/elbasvir (NS5A inhibitor). The presently available therapies end in a cure for the majority of patients (over 95%), regardless of age, sex, race, body weight, severity of liver disease, comorbidities, or HIV or HBV coinfection [15,16,17,18].

The aim of our study was to evaluate the effectiveness and safety of antiviral DAA therapy in patients chronically infected with HCV with concomitant epilepsy.

## 2. Materials and Methods

The observation covered 184 epilepsy patients with chronic HCV infection (sample group) that received treatment between 2015 and 2018. The gathered data were compared to the results observed in a group of 3573 HCV-infected patients without comorbidities, except for HIV and/or HBV coinfection (control group). Both groups of patients were selected from a total of 10,152 patients in the EpiTer-2 database. This database was created under the auspices of the Polish Association of Epidemiologists and Infectiologists and contains data received from 22 national hepatological centers in Poland. These data cover the methods, effectiveness, and safety of the treatment of Polish HCV-infected patients with liver disease at varying degrees of advancement in actual clinical practice. All patients were treated in accordance with the current recommendations of the National Health Fund and the Polish Group of HCV Experts [14]. Each of the HCV-infected epilepsy patients underwent a neurological examination before starting the treatment. The choice of anti-HCV therapy was made in accordance with the SmPC guidelines for a given type of drug, taking into account possible drug interactions with the anti-epileptic drugs already in use (or changing the anti-epileptic drugs, with the patient’s consent), ARV drugs, anti-HBV drugs, and, if they were used, hormonal contraceptives. When deemed necessary, the anti-epileptic treatment was modified. The most common modification was the discontinuation of carbamazepine and phenytoin.

### 2.1. Study Procedures

The selection criteria for both groups were aimed at adults with confirmed HCV infection (positive results for anti-HCV antibodies and HCV viremia) and defined staging of liver fibrosis. In the epilepsy group, an epilepsy diagnosis was confirmed by a neurologist, and each patient underwent additional tests, including an EEG, and continued anti-epileptic therapy. The control group consisted of generally healthy people—not counting the HCV infection—without comorbidities.

Patient demographics and information on comorbidities, treatment, the stage of liver disease, the activity of the inflammatory process in the liver, HBV and HCV coinfections, and the HCV genotype were assessed at baseline. HCV RNA VL was assessed at baseline, at the end of treatment (EOT), and 12 weeks after the end of SVR treatment (Table 2).

### 2.2. Assessment of Liver Disease Severity

The fibrosis stage was assessed noninvasively (intermittent histology testing) using transient elastography (Fibroscan, Echosens Paris, France), or shear-wave elastography (Aixplorer Supersonic Imagine France). Each patient was examined using only one of the above methods. The cut-offs for the prediction of F4 fibrosis used with both types of elastography assessment were established at the level of 13 kiloPascals, based on the recommendations of the EASL (European Association for the Study of the Liver). All patients with liver cirrhosis were also classified using the Child–Turcotte–Pugh score.

### 2.3. Statistical Analysis 

The results were expressed as means ± standard deviations or *n* (%). *p* values of <0.05 were considered statistically significant. Comparisons between groups were performed through the analysis of non-parametric tests. The significance of differences was calculated using Fisher’s exact test or a chi-square test (for appropriate, nominal variables), and the Mann–Whitney U test was used for continuous variables. Statistical analyses were performed using GraphPad Prism 5.19 (GraphPad Software, Inc., La Jolla, CA, USA).

## 3. Results

HCV-infected patients with coexisting epilepsy accounted for over 1.8% of all HCV-infected patients in the Epiter-2 database. Patient demographics and information on comorbidities, types of treatment, the stage of liver disease, the activity of the inflammatory process in the liver, HBV and HCV coinfections, HCV genotype, and baseline viremia are summarized in Table 2. Compared to the control group, the sample group stood out, with a statistically significantly greater number of women, a higher patient BMI, more frequent HBV coinfections, and more frequent use of additional medications. Patients with epilepsy also had a statistically significantly higher HCV VL at the beginning of treatment, a slightly higher intensity of liver fibrosis, and a lower number of peripheral blood platelets.

Data on the methods of pharmacotherapy chosen to manage the HCV infections in the sample group are summarized in Table 3, which reflects the progress in the access to more modern DAA therapies observed between 2015 and the end of 2018. The access of HCV therapy patients without epilepsy changed over time in a similar manner. In accordance with the recommended standards, the duration of treatment in both groups ranged from 12 to 24 weeks, depending on the treatment regimen, drug availability, the amount of viral load, and the degree of liver disease. After the initial correction of possible drug interactions between antiviral drugs and anti-epileptic medications, drug interactions and, consequently, the need to modify treatments occurred in only two people. Moreover, the observed side effects of DAA therapy in the sample group were mild and occurred in a smaller percentage than in the control group patients (Table 4). The effectiveness of anti-HCV treatment in the sample and control groups was high and did not show a statistically significant difference, regardless of the genotype or stage of liver disease: SVR in the sample group, with ITT = 93.5% and mITT = 95.5%, and the control group, with ITT = 95.2% and mITT = 97.5% (Table 5).

No serious adverse events were reported in either the sample group or the control group, except for the one described below in the epilepsy group. In both groups, mild gastrointestinal symptoms, headaches, and sleep issues were similarly frequent. 

Twelve patients who did not complete the treatment, did not respond to HCV-NR therapy, or relapsed (Table 6) were analyzed separately with a focus on factors that could affect the final response to an antiviral treatment (Table 6). The death of one patient with epilepsy was not related to the therapy and was a typical complication of advanced cirrhosis of the liver complicated by portal hypertension (gastrointestinal hemorrhage). Three patients discontinued treatment without justification (these were patients with additional issues: HIV coinfection, alcoholism, and depression). The remaining patients in which the antiviral treatment turned out to be ineffective were almost exclusively male, were significantly older than those who responded to treatment, had higher viral loads, were infected with the 3HCV genotype (50% of the group), and had clinically diagnosed cirrhosis. Additionally, laboratory tests in this group revealed lower numbers of platelets and lower concentrations of serum albumin.

## 4. Discussion

We observed that the percentage of epileptic patients among HCV-infected patients was almost two times higher than the percentage of epileptic patients in the general Polish population (approximately 1%). This is probably due to the fact that epileptic patients come in contact with health care much more frequently, which is conducive to infection.

Several studies have shown that HCV infections are related to some neuropsychiatric conditions (i.e., extrahepatic HCV manifestations (EHM)), increasing the risk of their occurrence by more than two times. This includes mood disorders, depression, cognitive dysfunction, sleep disorders, and chronic fatigue syndrome [7]. Some also suggest that HCV may take part in demyelination and in functional and organic disorders of the brain and may be related to certain forms of encephalitis [19]. To date, however, no relationship between HCV infection and any form of epilepsy has been confirmed. However, it is observed that patients with epilepsy of various etiologies require medical attention more frequently [20], experience more injuries (due to epileptic seizures, coexisting alcoholism, and self-mutilation), and exhibit risky behaviors—all of which potentially increase the risk of being infected with hepatotropic viruses [21,22]. This was reflected in our observations: the number of active HCV infections was two–four times higher in epilepsy patients than in the entire HCV-infected population. Other researchers also observed a higher incidence of HCV infection among epilepsy patients [23]. This points to the need for the epidemiological surveillance of all epilepsy patients and the monitoring of this group for hepatotropic virus infections.

Compared to the control group patients (though the difference was statistically insignificant), the sample group patients were characterized by more advanced liver disease and a larger number of known potential factors for worse responses to treatment (numerous comorbidities, HBV coinfection, higher BMI, and numerous comedications). This, however, did not translate into a drop in the effectiveness of antiviral treatment in this group of patients. What it may suggest is the delayed diagnosis of liver disease, the focus of neurology specialists on epilepsy therapy, and/or the disregard for health issues other than epilepsy [24,25]. Epilepsy patients diagnosed with HCV infection before the DAA era were most often not eligible for treatment due to the numerous interactions and side effects of interferon. Even today, with DAAs available, before starting anti-HCV treatment patients require medication adjustments (in accordance with the SmPC) due to potential drug interactions; this eventually leads to changing the anti-epileptic therapy. The effectiveness of anti-HCV therapy based on various treatment regimens (most of them are now considered outdated) was high in the ITT group (93.5%) and in the mITT group (95.5%), similar to the results obtained in the control group, i.e., patients without comorbidities, and consistent with the findings observed by other research teams on the topic of the effectiveness of antiviral DAA treatment [26,27].

The treatment of 20 patients from the sample group conducted in 2016 using the now-discontinued therapeutic regimens that are regarded to be less effective, i.e., Sofosbuvir/Ribavirin [28], Sofosbuvir/Pegylated -interferon alfa 2a + -Ribavirin, Sofosbuvir/Simeprevir, and Asunaprevir/Daclatasvir, undoubtedly stands behind the statistically insignificant drop in effectiveness observed in the sample group and cited research sources [29]. The group of eight patients who did not respond to antiviral therapy was characterized by the accumulation of recognized factors for worse responses, which included male gender, advanced age, cirrhosis, high HCV replication, and genotype 3 infection.

Anticonvulsant therapies can affect the course of liver disease. This is especially true for older drugs such as valproic acid, phenytoin, and carbamazepine [30,31,32]. These drugs are metabolized by the liver and may have a toxic effect. The additional presence of infection significantly increases the risk of disease progression and fibrosis in this group of patients; hence, the elimination of this risk is very important. There are also drugs that are known to imitate advanced liver disease, such as benzodiazepines and valproic acid; they may cause hyperammonemia and encephalopathy, yet do not have a permanent damaging effect on the liver [33].

As many as three patients with alcoholism, depression, or HIV coinfection co-occurring with epilepsy discontinued treatment without justification, which may suggest that a more detailed qualification process is necessary for the assessment of HCV therapy candidates in the context of proper cooperation with the patient.

## 5. Conclusions

In actual clinical practice, the therapy of HCV infection using DAA drugs in patients suffering from and treated due to epilepsy is highly effective. This effectiveness is comparable to the effectiveness of the treatment of patients with no significant comorbidities.The careful qualification for antiviral treatment and the correction of anti-epileptic treatments, in line with SmPC guidelines, that are performed before starting antiviral therapy determine the safety and effectiveness of treatment, preventing significant drug interactions between antiviral and anti-epileptic drugs.Before starting the undoubtedly expensive DAA therapy and in order to avoid the unjustified discontinuation of treatment and possible adverse drug interactions, HCV-infected patients with epilepsy of various etiologies need to be carefully assessed to check whether they understand the principles of the therapy and the need for systematic monitoring of the treatment.

## Figures and Tables

**Table 1 jpm-13-01111-t001:** Drug interactions between anti-epileptic drugs and the currently available HCV DAAs (EASL 2020).

Drug	SOF	SOF/VEL	SOF/VEL/VOX	GLE/PIB	GZR/EBR
Carbamazepine					
Clonazepam	low	low	low	low	low
Eslicarbasepine	low				
Ethosuxcimide	low	low	low	low	low
Gabapentin	low	low	low	low	low
Lacosamide	low	low	low	low	low
Lamotrigine	low	low	low	low	low
Levetiracetam	low	low	low	low	low
Lorazepam	low	low	low	low	low
Oxcarbazepine					
Phenobarbital					
Phenytoin					
Primidone					
Topiramate	low	low	low	low	low
Walproate	low	low	low	low	low
Zonisamide	low	low	low	low	low

SOF: sofosbuvir, SOF/VEL: sofosbuvir/velpatasvir, SOF/VEL/VOX: sofosbuvir/velpatasvir/voxilaprevir, GLE/PIB: glecaprevir/pibrentasvir, GZB/EBR: grazoprevir/elbasvir. High-blue pluse means a high risk of interaction between drugs. Low means a low risk of interaction between the epileptic and study drugs.

**Table 2 jpm-13-01111-t002:** Baseline characteristics of sample group patients (ITT).

Parameter	HCV/Epilepsy N = 184	HCV N = 3573	*p*-Value
Sex: female/male	59 (32%)/125 (68%)	1719 (48%)/1854 (52%)	<0.001
Age (years), mean ± SD (min–max)	46.6 ± 13.6 (20–81)	44.7 ± 12.9 (18–91)	0.06
BMI, mean ± SD (min–max)	26.1 ± 4.3 (16.9–40.6)	25.2 ± 3.9 (14.6–56.5)	0.007
Any comorbidity (*n*)	179 (97%)	0 *	N/A
Hypertension (*n* (%))	55 (30%)	0	N/A
Diabetes	12 (7%)	0	N/A
Kidney disease	7 (4%)	0	N/A
Autoimmune disease	2 (1%)	0	N/A
HCC and non-HCC npl	4 (2%)	0	N/A
Coinfections (%)			
HBV	29 (16%)	380 (11%)	0.04
HIV	11 (6%)	279 (8%)	0.44
Concomitant medications (%)	167 (91%)	607 ** (17%)	<0.001
HCV genotype (*n* (%))			
1	5 (3%)	66 (2%)	
1a	9 (5%)	190 (5%)	
1b	138 (75%)	2670 (75%)	0.83
2	0	5 (0.1%)	
3	25 (13%)	447 (13%)	
4	7 (4%)	195 (5%)	
HCV viral load >10^6^ (*n* (%) IU/mL)	106	1686	0.007
mean ± SD	3,060,000 ± 6,269,000	2,268,000 ± 5,878,000	0.04
Liver fibrosis (*n* (%))			
F0	0	82 (2%)	
F1	63 (34%)	1669 (47%)	
F2	32 (17%)	705 (20%)	<0.001
F3	40 (22%)	437 (12%)	
F4	45 (25%)	609 (17%)	
No data	4 (2%)	71 (2%)	
CTP (*n* (%))			
A	175 (95%)	3415 (96%)	
B	4 (2%)	71 (2%)	0.90
C	0	6 (0.2%)	
No data	5 (3%)	78 (2%)	
Bilirubin (mg/mL), mean ± SD	0.74 ± 0.52	0.78 ± 0.61	0.37
ALT (IU/mL), mean ± SD	73.4 ± 46.1	77.5 ± 64.2	0.41
INR, mean ± SD	1.1 ± 0.3	1.1 ± 0.2	0.14
Creatinine (mg/mL), mean ± SD	0.82 ± 0.23	0.80 ± 0.17	0.53
Hemoglobin (g/dL), mean ± SD	14.7 ± 1.4	14.6 ± 1.6	0.76
Platelet count (×10^3^)/μL	176 ± 70	201 ± 72	<0.001

* Excluding HBV, HIV, and liver cirrhosis with related complications. ** Including medications for HBV, HIV, liver cirrhosis decompensation and other complications, herbal hepatoprotectives, oral contraceptives, and other preventive medications.

**Table 3 jpm-13-01111-t003:** Anti-HCV treatment regimens in HCV-infected patients with epilepsy (*n* = 184).

Regimens	No. of Patients (%)
Sofosbuvir/Ribavirin	17 (9)
Sofosbuvir/Pegylated-interferon alfa 2a ± Ribavirin	1 (0.5)
Sofosbuvir/Simeprevir	1 (0.5)
Asunaprevir/Daclatasvir	1 (0.5)
Ledipasvir/Sofosbuvir	51 (27.7)
Ledipasvir/Sofosbuvir/Ribavirin	20 (10.9)
Elbasvir/Grazoprevir	41 (22.3)
Ombitasvir/Paritaprevir/Ritonavir/Dasabuvir	27 (14.7)
Ombitasvir/Paritaprevir/Ritonavir/Dasabuvir/Ribavirin	9 (4.8)
Ombitasvir/Paritaprevir/Ritonavir/Ribavirin	3 (1.6)
Sofosbuvir/Velpatasvir	7 (3.8)
Sofosbuvir/Velpatasvir/Ribavirin	1 (0.5)
Glecaprevir/Pibrentasivir	5 (2.7)

**Table 4 jpm-13-01111-t004:** Summary of adverse events in two groups (patients with HCV/epilepsy and patients with HCV without comorbidities), with data on treatment course, modifications, discontinuation, and safety (ITT).

Parameter No. (%)	HCV/Epilepsy *n* = 184	HCV without Comorbidities (*n* = 3573)
DAA therapy modifications	2 (1%)	73 (2%)
DAA therapy discontinuation	3 (2%)	24 (0.7%)
Patients with any AE	51 (28%)	718 (20%)
Epilepsy aggravation	None	N/A
Death during DAA treatment	1 (0.5%)	7 (0.2%)

**Table 5 jpm-13-01111-t005:** Characteristics of HCV/epilepsy patients who failed to achieve SVR12 (sustained viral response after 12 weeks). LTFU—lost to follow-up, ARV—antiretroviral.

No.	Sex	Age	HCV Genotype	DAA	Treatment	Fibrosis	Epilepsy	BMI	ALAT	CPT	Treatment Modifications	Liver Decompensation	Epilepsy Episode during Treatment	Type of Non-Response
				Regimen	Duration		Therapy							
1	M	39	3	SOF/PEGIFNalfa/RBV	12	4	amizepinCR	33	77	A	no	no	no	Relapser
2	M	59	1b	LDV/SOF	12	4	lamotrigin		55	A	no	no	no	NR
3	M	63	1b	LDV/SOF	12	4	levitiracetam	N/A	53	A	no	no	no	Relapser
4	F	38	1b	GZR/EBR	12	3	without epi treatment, alcoholic	23	123	-	no	no	no	NR
5	M	54	3	SOF/RBV	24	4	valproid acid, levitiracetam	29	79	A	no	no	no	Relapser
6	M	57	1b	OBV/PTV/r+DSV+RBV	2	4	levitiracetam	Na	94	A	2-week therapy interrupted by GI hemorrhage	yes	no	Death
7	M	61	1b	GZR/EBR	12	4	levitiracetam, sertraline	23	45	A	no	no	no	LTFU
8	M	57	3	GLE/PIB	16	4	levitiracetam	33	94	A	no	no	no	Relapser
9	M	50	3	SOF/RBV	24	4	without epi treatment	25	117	A	no	no	no	Relapser
10	M	31	3	SOF/RBV	2	N/A	valproid acid, diazepam, trazodon, ARV, alcoholic	18	150	-	2-week therapy interrupted on patient’s demand	no	no	LTFU
11	M	75	1b	LDV/SOF	12	3	valproid acid	25	37	-	no	no	no	Relapser
12	M	40	3	VEL/SOF	12	2	trazodone, clonazepam, ARV	22	16	-	no	no	no	LTFU

**Table 6 jpm-13-01111-t006:** Factors influencing the response to DAA treatment in HCV/epilepsy patients (mITT—modified intent to treat).

Parameter	SVR (*n* = 172)	Non-SVR (*n* = 8)	
Sex, female/male	58 (34%)/114 (66%)	1 (12%)/7 (88%)	<0.001
Age (years), mean ± SD (min–max)	46 ± 13 (20–81)	57 ± 10 (39–75)	0.03
BMI, mean ± SD (min–max)	26.1 ± 4.2	27.7 ± 4.4	0.41
Any comorbidity	172 (100%)	6 (75%)	NA
Hypertension	53 (31%)	2 (25%)	NA
Diabetes	11 (6%)	1 (12.5%)	NA
Kidney disease	7 (4%)	0	NA
Autoimmune disease	2 (1%)	0	NA
HCC and non-HCC npl	4 (2%)	0	NA
Others Coinfections (%)			
HBV	25 (15%)	2 (25%)	0.34
HIV	9 (5%)	0	NA
Concomitant medications (%)	155 (90%)	8 (100%)	1.00
HCV genotype (*n* (%))			
HCV-GT1	146 (85%)	4 (50%)	
3	19 (11%)	4 (50%)	0.005
4	7 (4%)	0	
HCV viral load (IU/mL)	2,938,000 ± 6,268,000	4,799,000 ± 6,423,000	0.16
Liver fibrosis (*n* (%))			
F1	63 (37%)	0	
F2	31 (18%)	0	0.004
F3	38 (22%)	1 (12.5%)	
F4	38 (22%)	6 (75%)	
No data	2 (1%)	0	
Bilirubin (mg/mL), mean ± SD	0.73 ± 0.51	0.88 ± 0.36	0.16
ALAT (IU/mL), mean ± SD	72.5 ± 45.7	69.6 ± 27.0	0.79
Albumin (g/dL), mean ± SD	4.2 ± 2.2	3.6 ± 0.6	0.03
Creatinine (mg/mL), mean ± SD	0.82 ± 0.24	0.76 ± 0.19	0.43
Hemoglobin (g/dL), mean ± SD	14.7 ± 1.4	14.9 ± 1.2	0.78
Platelet count (×10^3^)/μL	177 ± 69	142 ± 62	0.11

## Data Availability

Data are placed in National Polish Data Base Epiter-2.

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
