# Peer review of "Real-World Effectiveness and Safety of Direct-Acting Antivirals in Patients with Chronic Hepatitis C and Epilepsy: An Epi-Ter-2 Study in Poland"

_jpm, 2023, doi:10.3390/jpm13071111_

Round 1

Reviewer 1 Report

In their manuscript, Pazgan-Simon with coauthors present data on the safety and effectiveness of DAA-based therapy of chronic HCV infection in patients with coexisting epilepsy. The data on HCV treatment is such population are scarce and, thus, have a significant value. The study is performed well but I have some specific comments as below.

1. Lines 67-73. The sentence is to long and, due to this, is hard to follow. It is recommended to split this sentence into several ones.

2. Table 1. Please provide the reference to Table 1 in the text and the explanation of the symbols and the color identifications used in the table. The image resolution is poor.

3. Please provide inclusion and exclusion criteria for both study and control group. Please specify the exact criteria for selection patients in the control group.

4. Table 2: Data in columns 2 and 3 are presented only as absolute numbers. However, in column 1 several parameters are given as n (%) or % only.

5. The content of Table 4 duplicates the content of Table 3. The table with data on SVR in patient groups (referenced in the text as Table 5) is missing.

6. Tables 3-5: data in % are not provided

7. Please provide the explanation for abbreviations used in the Table 6 (NR, LTFU, etc.).

8. Please specify, if patients who failed to response to DAA treatment were tested for the HCV resistance associated variants? Were they offered another treatment regimen?

9. Lines 208-210. The sentence is unclear. How HCV prevalence could be higher in epilepsy patients compared to total HCV-infected population? Do authors mean the proportion of viremic cases (active infections) in two anti-HCV reactive cohorts? Even if so, difference in proportion of viremic patients not necessarily reflects the higher rates of HCV exposure and higher risk of infection in patients with epilepsy. To draw such a conclusion, data on HCV prevalence in patients with epilepsy should be compared to that in general population. Instead, higher proportion of viremic patients (data on this topic are not shown in this study) and higher viral load in epilepsy (shown in this study) may be indicative of lower immune control of HCV infection in epilepsy patients.

10. Lines 210-211. “Other researchers also observed a higher incidence of HCV infection 210 among epilepsy patients” - Reference is needed.

11. Please provide in Discussion any available data whether anti-epileptic drugs are hepatotoxic and could worsen the course of untreated HCV infection?

In general, English language is fine. However, some sentences are too long to follow and will benefit from splitting into several ones.

Author Response

Thank you kindly for your comments on the article.

Here are my replies and corrections:

  1. The sentence in question has been divided into three separates ones, making it more precise.
  2. I added references to the table and changed it as per recommendation given.
  3. I presented the selection criteria for the control and study groups.
  4. In table 2, I added the results in percentages.
  5. Data corrected.
  6. Percentages have been complemented.
  7. Explanations of abbreviations have been added.
  8. Patients were not tested for resistance to therapy, this was not the aim of our work, only the response to the first therapy.
  9. The sentence in question has been changed.
  10. The reference has been added, although it concerns a different study. Epidemic assessment of HCV in people with epilepsy is rare.
  11. A paragraph on the hepatotoxicity of anticonvulsants and their effect on chronic liver disease has been added in the discussion section.

Thank you

Reviewer 2 Report

General Comments: The manuscript titled "Real-world effectiveness and safety of direct-acting antivirals in patients with chronic hepatitis C and epilepsy: an Epi-Ter-2 study in Poland" investigates the effectiveness and safety of HCV treatment in patients with coexisting epilepsy. Overall, the study addresses an important research question; however, several areas require attention and improvement. The comments below outline specific concerns and suggestions to enhance the quality, validity, and interpretability of the manuscript.                                                                                                            Comment 1: Specify the sources of data for the prevalence of HCV infection and epilepsy in Poland.

Comment 2: Provide more specific information about the antiviral therapies used in the study.

Comment 3: Elaborate on the study's aims in the abstract.

Comment 4: Provide more details on the patient selection process and criteria for defining the epilepsy and control groups.

Comment 5: Include specific numerical results to support the claims about the effectiveness and safety of HCV treatment in patients with epilepsy.

Comment 6: Provide detailed information on the patient selection process to address potential selection bias.

Comment 7: Include a comprehensive analysis of adverse events experienced by patients in both epilepsy and control groups.

Comment 8: Describe the nature and extent of modifications to anti-epileptic treatment and their implications on treatment effectiveness and safety.

Comment 9: Specify the duration of follow-up after treatment completion to evaluate treatment outcomes and long-term safety.

Comment 10: Discuss potential confounding factors such as epilepsy types, severity, and anti-epileptic drug use.

Comment 11: Improve image quality or provide alternative methods for presenting data.

Comment 12: Proofread the manuscript for grammatical errors, improve clarity, and ensure the abstract concisely summarizes key findings, methods, and limitations.

The manuscript demonstrates satisfactory English language proficiency, but there are areas that require improvement. Attention to grammar, vocabulary, coherence, and clarity will enhance the overall quality of the writing.

Author Response

I greatly appreciate all your comments. I have made corrections to the text as per your recommendations.

  1. I added a source on the epidemiology of epilepsy in Poland.
  2. I added information on antiviral treatment.
  3. I added the objectives of the study in the abstract.
  4. I have supplemented the data of patients qualified for treatment and I have also added the diagnostic regimen used with epileptic patients.
  5. Data on treatment results of treatment have been supplemented.
  6. Data have been supplemented.
  7. Side effects were rare in both groups, information in the text has been supplemented.
  8. The few modifications in anticonvulsant therapies involved discontinuation of older drugs and have been included in the text.
  9. The follow-up period after treatment was 6 months and was standard for all patients.
  10. Unfortunately, I do not possess data on different types of epilepsy, the severity of the disease and the specific aspects of treatment in this group of patients.
  11. Hopefully, I have improved the quality of the presentation.
  12. Grammar and spelling were checked.

Thank you again for your comments.

Round 2

Reviewer 2 Report

Dear Authors,

I have carefully reviewed the revised version of your manuscript titled "Real-world effectiveness and safety of direct-acting antivirals in patients with chronic hepatitis C and epilepsy: an Epi-Ter-2 study in Poland." I would like to commend you on the thorough revision and improvements made, particularly in terms of language and addressing other major concerns raised during the previous review.

The clarity and coherence of the manuscript have significantly improved. The revised text demonstrates a strong command of English, making it easier for readers to follow and understand your study. The modifications made to the language and structure have resulted in a more polished and professional manuscript.

I also appreciate the attention given to other major concerns highlighted in the previous review. You have effectively addressed these concerns and provided additional clarification where needed. The revisions have greatly enhanced the overall quality and scientific rigor of your work.

I would like to commend you on your diligence in incorporating the feedback and suggestions provided. It is evident that you have carefully considered the reviewer's comments and made the necessary revisions to strengthen your study.

Considering the thorough revision, improved language, and addressing of major concerns, I recommend accepting the revised version for publication. The manuscript now meets the standards required for a final draft publication.

Once again, I commend you on the significant improvements made, and I appreciate your efforts in ensuring the manuscript's readiness for publication.

NA